# Factors involved in vulvar pain during sexual activity and persistence in sexual activity amidst pain

Carlotta Oesterling[1]*, Amelie Harder[2], Charmaine Borg[1‡], Peter de Jong[1‡]

**1** University of Groningen, Groningen, the Netherlands, **2** University of Oxford, Oxford, United Kingdom

‡ CB and PDJ are joint senior authors.
* c.f.oesterling@rug.nl

## Abstract

Evidence suggests that women often endure pain during sexual activity, continue engaging in such activity despite experiencing pain, and tend to avoid communicating these painful experiences to their partners. The present study aims to shed light on psychosocial factors that may contribute to vulvar pain and the engagement in sexual activity despite pain, with a specific focus on the relevance of sexual self-esteem, the definition of sex (limited to penile-vaginal intercourse or inclusive of other intimate behaviours), sexual agency, and sexual motivation. A sample of N = 277 female students of a Dutch University was included. Participants were between 18 and 33 years old. The primary outcome measures were female sexual distress, sexual function, and vulvar pain. Engagement in sexual activity despite pain, pain communication, sexual agency, and relationship satisfaction were included as mediators. The majority of participants (80.0%) reported to experience pain at least sometimes, and 15.0% reported to experience pain more than half of the time. Engaging in penile-vaginal intercourse despite experiencing pain was common, with 42.0% of participants indicating to do so always or most of the time and 65.0% at least sometimes when experiencing pain. Of the affected women, 41.0% did not communicate pain to their partners. Low sexual self-esteem, a restrictive definition of sex, limited sexual agency, and low autonomous sexual motivation were all significantly related to at least one of the primary outcome variables. These associations were partly mediated by engagement in PVI despite pain, (no) pain communication, and (low) relationship satisfaction.

## Introduction

Vulvar pain during sexual intercourse is highly prevalent in women, with 10.0–28.0% of women in the general population reporting persistent pain during sexual intercourse [1]. Vulvar pain can be associated with psychosocial (i.e., intimacy, sexual communication and affection, childhood maltreatment, anxiety and depression,

**Data availability statement:** The data is now accessible at dataverse.nl (https://dataverse.nl/dataset.xhtml?persistentId=-doi:10.34894/ESGWZO), with the DOI https://doi.org/10.34894/ESGWZO. Data reference: Oesterling, Carlotta; Amelie Harder; Borg, Charmaine; de Jong, Peter, 2025, "Factors Involved in Vulvar Pain during Sexual Activity and Persistence in Sexual Activity amidst Pain", https://doi.org/10.34894/ESGWZO, DataverseNL, V1.

**Funding:** The author(s) received no specific funding for this work.

**Competing interests:** The authors have declared that no competing interests exist.

partner responses to pain, pain catastrophizing and self-efficacy, and sexual motivation) and biomedical factors (i.e., co-morbidities with other functional pain syndromes, genetics, hormonal factors, inflammation, musculoskeletal and neurologic mechanisms [1]). Pain symptoms persisting for over six months and causing significant sexual distress are classified as Genito-Pelvic Pain/Penetration Disorder [2], which includes vulvar pain during penetrative intercourse caused by involuntary contraction of the pelvic floor muscles upon attempted vaginal entry (vaginismus), pain (dyspareunia) localized to the vestibule of the vagina (provoked vestibulodynia), or at other vulvovaginal locations (vulvodynia), and fear and anxiety elicited by attempted penetration [2], or by merely the idea of penetration [3].

Experiencing vulvar pain during penetration can come with significant sexual distress and may negatively impact sexual arousal, desire, and lubrication. In light thereof, it is striking that research found a substantial proportion of heterosexual women affected by vulvar pain to proceed engaging in penile-vaginal intercourse (PVI) [4,5], with only 50.0% of women effectively communicating the experience of pain to their partner [6,7]. A representative survey study has further found that 60.0% of affected women indicated experiencing the motivation to continue engaging in PVI despite pain [6]. While – given the reduced focus on penetration – it is likely that these numbers are lower in non-heterosexual female couples, corresponding statistics have not yet been established.

Against the backdrop of these findings, the aim of the present study was twofold. The first aim was to test the robustness of the finding that a large proportion of women engages in sexual intercourse despite pain and avoids communicating respective pain to their sexual partners. By replicating the design applied by Carter et al. [6], it was examined to which extent women engage in sexual intercourse despite pain, whether they communicate the experience of pain to their partners, and which reasons may underlie the motivation not to do so. Second, it was aimed to improve insight in the factors that contribute to vulvar pain and women's persistence in sexual intercourse despite experiencing pain. On the basis of the available findings in the literature, the following candidate factors were included: (low) sexual self-esteem, the definition of "having sex" being limited to penetrative intercourse, (low) sexual agency, and autonomous sexual motivation. A comprehensive heuristic model (see Fig 1) was taken as a starting point to test the relevance of these factors for explaining vulvar pain, problems in sexual functioning, and sexual distress in women. Based on this model, it was further explored in how far the candidate factors are related to women engaging in penetrative intercourse despite pain, as well as not communicating the experience of pain to their partners. The heuristic model integrates findings of the literature on each of the distinct constructs and offers a novel approach to capturing how several psychosocial factors may influence each other, the engagement in PVI despite pain, and sexual pain, functioning and distress.

## Sexual scripts and a restrictive definition of sex

Within the context of heterosexual relationships, both men and women follow sexual scripts directing and informing their sexual behaviour. Despite increasingly egalitarian

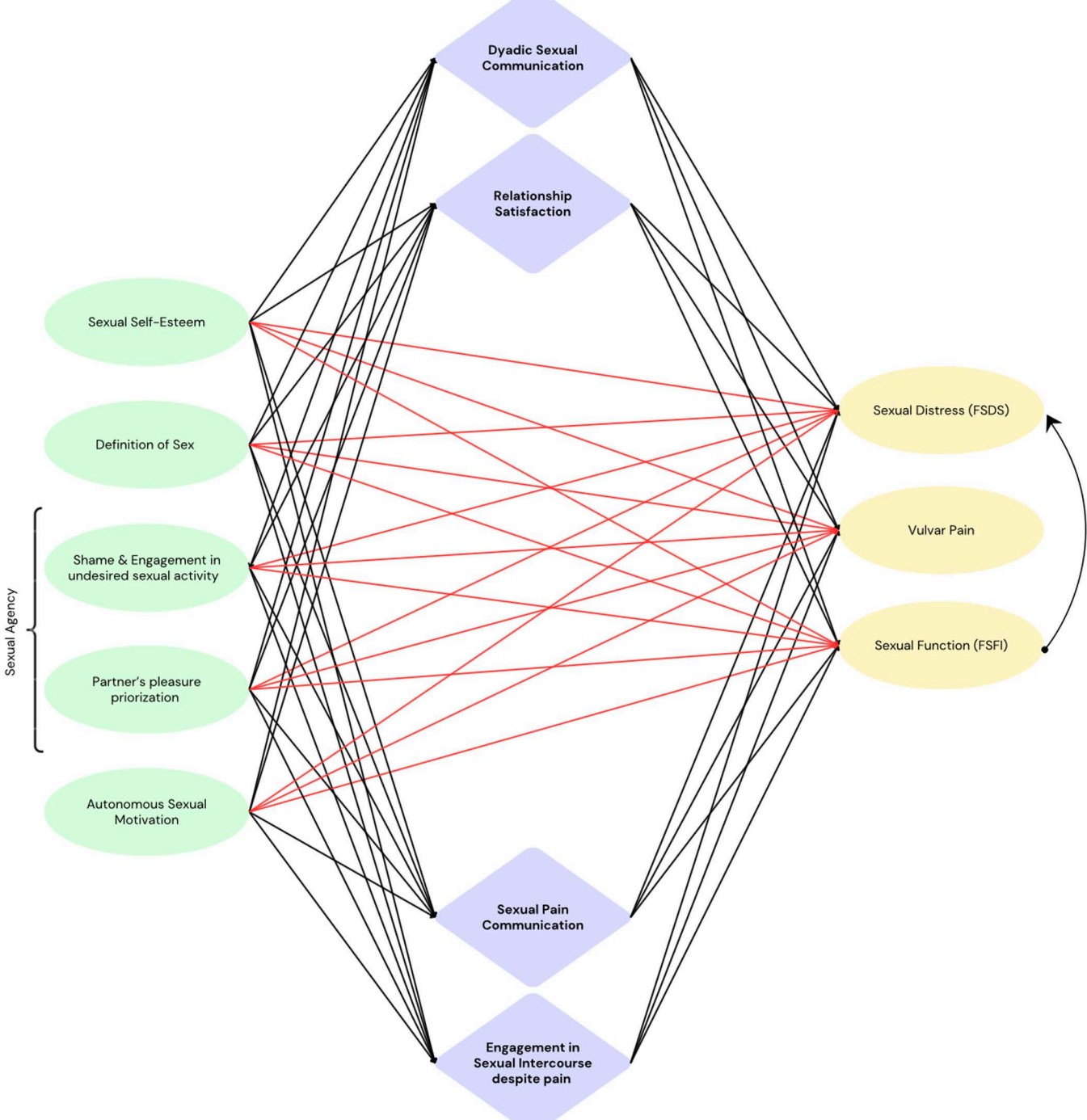

**Fig 1. Heuristic Model.** Note. Green: Observed predictor variables (model allows for covariance among predictor variables). Lilac: Observed mediator variables. Yellow: Observed outcome variables. Red arrows: Direct effects. Black arrows: Indirect effects. Predictors were allowed to covary.*Following behaviours were presented as response options: penile-vaginal intercourse (with and without orgasm), oral sex (fellatio), oral sex (cunnilingus), giving anal sex, receiving anal sex, kissing, mutual masturbation, masturbation, intimacy and physical touch without penetration.

roles of men and women in heterosexual relationships, sexual scripts have been found to remain relatively stable in short- and long-term relationships [8,9], reinforcing male agency and female receptivity [9,10]. Adherence to traditional scripts offers guidance, predictability, and clarity but may carry potential drawbacks for both women and men. Women may experience discomfort initiating sexual activity, feel responsible to hold the role of the "gate-keeper" by delaying the first sexual encounter to occur with a new partner, and limit the expression of their sexual desires and boundaries [11–13].

For men, script adherence may lead to performance pressure and restricted emotional expression. The majority of studies on Sexual Scripting Theory (SST) [14], which postulates that cultural scenarios, informed by shared cultural norms and values, deeply influence personal motives and motivations as well as intrapersonal scripts [10,14,15], have been published prior to 2015. Although studies on SST thus precede recent social changes regarding gender norms and equality and the rethinking of concepts of femininity and masculinity, SST underscores that sexual behaviour and sexual expectations are deeply rooted in our cultural and social background and upbringing. The consequential deep embedding of values may partially explain why PVI remains the epitome of "normal" heterosexual sex [16,17] and women often continue to act less sexually agent than men in spite of normative changes in heterosexual relationships – which may contribute to the persisting gap in orgasm and pleasure experience between men and women that has been observed continuously in research [18,19].

Although research finds men of all sexual orientations to be more likely to achieve an orgasm than women, the largest difference is continuously found between heterosexual men, of whom 95.0% experience an orgasm usually or always, and heterosexual women, of whom only 65.0% experience an orgasm usually or always [20–25]. PVI only results in orgasm in 25% - 30% of women, while 95.0% of men experience orgasm during PVI. In contrast, women who have sex with women experience an orgasm in 80.0–90.0% of the time [20] – illustrating that the widely held notion of women being biologically less prone to orgasm stems from an oversimplification of female anatomy and response [18]. Therefore, the orgasm gap may rather emanate from the pervasive influence of social norms and the prioritization of PVI in heterosexual couples. Orgasm may thus frequently not be expected by women engaging in heterosexual intercourse, and its absence is consequently not evaluated as deprivation [26], resulting in a discrepancy of orgasm (and consequential pleasure) expectancies between men and women. Germane to this, research has found that women's lower likelihood of experiencing physical pleasure and orgasm when engaging in heterosexual sex results in lower expectancies of pleasure and orgasm [27,28] and has found women to feel less entitled to their own pleasure, while prioritizing the relationship and their partner's pleasure [9,29–32]. The prioritization of the partner's pleasure over their own may further emanate from feelings of shame and guilt and a desire to align with one's femininity and sexual identity [33], which might be threatened by female sexual dysfunction in light of societal norms of implying PVI as "having sex" [34] – leaving women feeling unable to fulfil their own and their partners' expectations when refraining from penetrative sex when experiencing pain during PVI.

These commonly held notions and expectations about sexual activity and alignment to gender norms can have detrimental effects on health when it comes to vulvar pain and Genito-Pelvic Pain/Penetration Disorder. For instance, women may learn to endure higher levels of pain and discomfort without refraining from PVI. When anticipating and experiencing pain, the friction will further augment pain and tissue damage, increasing concomitant defensive responses (i.e., pelvic floor tension), low arousal, and pain expectancy and catastrophizing [35,36]. Further, it has been shown that women experiencing pain have more difficulties refusing undesired sex, feel more inferior to their partner, and place higher value on the partner's satisfaction than women without pain [29,33]. Moreover, women who continued to engage in sexual intercourse despite pain identified themselves as inadequate sexual partners, experiencing shame, guilt, and decreased desire.

In light of these findings, concern has been brought forward regarding the large proportion of women engaging in sexual intercourse despite pain. Consequentially, the need to promote the importance of women's own satisfaction, and the engagement in sexual activities besides penetrative sex is widely expressed in research and clinic [37]. As PVI remains central to the definition of heterosexual sex, the present research will test whether a restrictive definition of sex (that does not expand to other behaviours) is associated with higher sexual distress, and possibly also reduced sexual function and

increased vulvar pain, as not being able to engage in PVI may then easily be concomitant to shame, guilt, and unfulfilled expectations in heterosexual women and their partners. Thereby, it is predicted that a restrictive definition of sex may also foster engagement in sexual intercourse despite pain to avoid uncomfortable situations, conversations, and rejection. Contrarily, defining sex more broadly to include other activities would allow for engagement in behaviours that do not elicit pain but are still considered as "sexual activity". This is seen in homosexual women, and it can reduce sexual distress and foster engagement in unpainful and enjoyable intimate activities. Further, it is suggested that a broader definition of sex and a larger array of behaviours also comes with increased communication, making it easier to communicate pain while a restrictive definition of sex may be associated with women not communicating the pain to their partners.

## Sexual agency

Besides a restrictive definition of sex, there is ample evidence for sexual agency playing a significant role in sexual behaviour regarding initiation and communication of one's own sexual desires [26]. Low sexual agency can have negative consequences for heterosexual women and their partner's sexual satisfaction and may promote engagement in risky or undesired sexual behaviour, and as hypothesized in the present research, possibly also vulvar pain and sexual distress. Despite the existing consensus on the concept of sexual agency as the ability to form and communicate voluntary and informed decisions regarding one's sexual behaviour, there can be variations in its precise definition and the nuances attributed to it. As a result, the term sexual agency lacks a clear definition and a validated measurement tool; being used as synonym for sexual assertiveness and/or varying in broadness of the definition. In the present study, sexual agency is conceived of as "the ability to identify, communicate, and negotiate one's sexual needs, and to initiate behaviours that allow for the satisfaction of those needs" [26]. To test whether low sexual agency is associated with engaging in sexual intercourse despite pain and limited pain communication, as well as vulvar pain and sexual distress, a novel measurement tool yielding two separate dimensions of sexual agency (prioritization of the partner's pleasure & shame and engagement in undesired sexual activity) was utilized.

## Sexual self-esteem

Sexual self-esteem (SSE) refers to how individuals perceive themselves in terms of their sexuality. It includes aspects such as attractiveness, confidence, and self-acceptance, which are pivotal in promoting mental health and influencing one's sexual encounters [38]. Sexual self-esteem has been identified as important for sexual satisfaction, assertiveness, and entitlement for sexual pleasure [39]. Further, sexual self-esteem has been shown to positively impact sexual communication, allowing women to verbally and non-verbally communicate which sexual behaviours they find satisfying [39]. Therefore, it is hypothesized that lower sexual self-esteem will predict increased vulvar pain and sexual distress, and lower sexual function. It is also hypothesized that lower sexual self-esteem predicts engagement in sexual intercourse despite pain and not communicating pain to one's partner, which both are thought to mediate the relationships between sexual self-esteem and the other psychosocial factors (i.e., definition of sex, sexual agency, sexual motivation) and sexual distress, vulvar pain, and sexual function.

## Autonomous sexual motivation

Self-determination theory conceptualizes behavioural motives on a continuum from fully autonomous to fully externally controlled [40–42]. Previous research has shown that applying self-determination theory [40,42] to sexual behaviour [43,44] may allow to derive a deeper understanding of motives for sexual interactions, and possibly also its link to sexual distress and vulvar pain. It is postulated that if individuals perceive they have efficacy (competence, i.e., self-esteem), agency (autonomy), and that they are connected to others (relatedness), they have the basis for internally motivated sexual behaviour, such as the joy of the activity itself. Therefore, in the present research it is assumed that autonomous sexual motivation reduces vulvar pain, sexual distress, and engagement in sexual intercourse despite pain, while increasing sexual function and pain communication.

Culturally embedded beliefs and expectations regarding sexual activity and one's own sexual behaviour may be adjusted through open sexual communication between partners, which positively affects sexual functioning and reduces sexual distress. Nevertheless, sexual communication is frequently inhibited in couples affected by vulvodynia [45,46], allowing for women's mere assumptions about what the partner expects and perceives as pleasurable (e.g., sexual activity including penetration as a necessity) to contribute to women frequently pursuing sexual intercourse despite pain. Therefore, predicted relationships are hypothesized to be mediated by not only whether women engage in sexual intercourse despite pain, and whether they communicate the pain to their partners, but also by dyadic sexual communication and overall relationship satisfaction (see also Fig 1).

### The present study

Based on the existing literature, it is hypothesized that, in line with results of Carter et al. [6], a considerable proportion of women experiencing pain during sexual intercourse do not communicate the pain to their partners and engage in intercourse despite experiencing pain, for which potential reasons were examined using thematic analysis, i.e., by identifying and interpreting recurring patterns or themes within qualitative responses to gain insights into underlying motivations to not communicate pain. Building on previous research, the herein tested comprehensive model will integrate several psychosocial factors such as the definition of sex, sexual agency, sexual self-esteem, and autonomous sexual motivation, allowing to derive insights on how these factors relate to each other, and to vulvar pain, sexual distress, the engagement of sexual intercourse despite pain and communication of pain. Engagement of sexual intercourse despite pain, pain communication, as well as dyadic sexual communication and relationship satisfaction are expected to mediate the relationships between the tested psychosocial factors and vulvar pain and sexual distress. The hypothesized heuristic model therefore provides an integrative and novel approach to vulvar pain and sexual distress, accounting for the relatedness and interdependence of psychosocial constructs that have yet been treated as distinct. As the breadth of individuals' definition of sex has not yet been taken into consideration when studying psychosocial correlates of sexual pain, the model provides a substantiation and extension of existing research. Highlighting the relationships among respective constructs may point to promising targets for future psychosocial interventions to address vulvar pain and sexual distress.

## Method

### Participants

Participants were recruited via availability sampling using SONA, a participant recruitment-platform of the University of Groningen. Data collection took place in April and May 2023. Sexually active women over 18 years of age were included into the sample, which consisted of $N = 277$ women between 18 and 33. Exclusion criteria comprised being pregnant, and being diagnosed with a psychiatric condition. All participants received course-credits as compensation. Following exclusions due to missing data, $n = 232$ women with a mean age of $M = 20.04$ ($SD = 2.06$) were included into the analyses. Of the included women, $n = 171$ (61.7%) identified as predominantly heterosexual, $n = 4$ (1.4%) identified as predominantly homosexual, and $n = 52$ (18.8%) identified as bisexual. Two women indicated to have children. The questionnaire was made available via Qualtrics survey software (Qualtrics, 2014) and completed online. Prior to commencement, all participants had given their written informed consent to participate in the study, which had gained approval by the ethical committee of the faculty of Behavioural and Social Sciences (PSY-2223–0118).

### Materials and procedure

Upon giving informed consent, participants were presented with a short online information video on the purpose and procedure of the study, followed by the online survey comprising multiple validated as well as novel, self-constructed questionnaires – which took 20–30 minutes to complete.

**Descriptive variables.** Participants were asked to provide demographic information, age, gender identity, education, number of children, their sexual orientation, the gender identity of their sexual partner(s), their relationship status, if applicable the gender identity of their relationship partner(s), relationship length, their status of living (i.e., alone, with their partner, with others), their religion, and if applicable, the extent to which their religion has an impact on their life.

**Vulvar pain and engagement in sexual intercourse despite pain.** Four self-constructed items were made available to assess vulvar pain and engagement of sexual intercourse despite pain. The first item appraised whether physical discomfort or pain is normally experienced upon (attempted) penetration, which could be responded to on a 5-point Likert scale ranging from 0 (No) to 4 (Yes, always). The achieved score was included as outcome variable in the model alongside female sexual function index (FSFI) and female sexual distress scale (FSDS) scores. Second, participants could choose one or more possible options for why the sexual intercourse could have been hurtful. Presented options were based on the Hite Report of Female Sexuality [47] and included options such as limited arousal and lubrication, yeast infection, and an open response box to add unmentioned causes of pain. Third, participants were asked to rate their average pain during intercourse on a scale from (1) *no pain* to (10) *worst pain* as recommended for clinical assessments of pain intensity [48]. Fourth, on a scale from 0 (No) to 4 (Yes, always), participants indicated whether they engage in intercourse despite pain, which was included as mediating variable into the model.

**Sexual communication pain.** As a replication of Carter et al. [6], all participants were presented with three items assessing whether they experienced pain during their *most recent* sexual activity 1 (not at all painful) to 5 (extremely painful), whether they had disclosed their pain to their partner (yes/no), and, if applicable, why they had not done so. The latter was presented as an open question, giving women the opportunity to bring forward any kind of reason behind not communicating their pain.

**Relationship satisfaction.** To evaluate the perceived quality of the participants' romantic relationships, the Perceived Relationship Quality Components Inventory (PRQC) [49] was employed. This self-report questionnaire consists of 18 items and assesses multiple aspects of relationship quality, including intimacy, commitment, trust, and overall satisfaction. Participants were asked to indicate the extent of their agreement or disagreement with each statement on a 5-point Likert scale, ranging from 1 (strongly disagree) to 5 (strongly agree). Internal consistency was high, with Cronbach's α = .98 in the present study.

**Dyadic sexual communication scale.** To measure the sexual communication within romantic relationships, the Dyadic Sexual Communication Scale (DSCS) [50] was used. The DSCS is a tool designed to assess the quality and effectiveness of sexual communication between intimate partners. It consists of 13 items and has been found to have high internal consistency by previous research, with Cronbach's α = .81. In the present study, Cronbach's α was lower, with α = .62.

**Female sexual function.** To assess overall female sexual function, the female sexual function index was employed (FSFI) [51]. The FSFI is the most widely used screening tool and outcome measure of female sexual function [52] and consists of 19 self-report items measuring five domains, namely arousal, satisfaction, desire, pain, and lubrication. Items refer to sexual activity that has taken place within the last four weeks and are responded to on a 5-point Likert scale ranging from 1 to 5, with higher scores indicating greater levels of sexual functioning on the respective item. A cutoff score of < 26 indicates the presence of clinical sexual dysfunction [52]. A total score below Psychometric properties of the FSFI are excellent, with internal consistency scores of Cronbach's α = .97 in the present study.

**Female sexual distress.** As consensus exists that for female sexual dysfunction to meet diagnostic criteria, sexual distress arises concomitant to sexual dysfunction [2], sexual distress is herein assessed using the widely used revised version of the female sexual distress scale (FSDS-R) [53]. The FSDS – R consists of 13 items and responses are provided using a response scale ranging from 0 (never) to 4 (always). A cut-off score of ≥ 15 was validated to indicate the presence of sexually related distress in a female sample [53]. Internal consistency was shown to be high, with Cronbach's α = .94 in the present study.

**Sexual self-esteem.** The short form of the Sexual Self-Esteem Inventory for Women (SSEI-W-SF) assesses the affective responses to the self-reported appraisals of sexual thoughts, feelings, and behaviours [54], and comprises 35 items of the original 81-item version [55]. Each item pertains to one of five subscales (skill/experience, attractiveness, control, moral judgement, adaptiveness) and can be responded to on a 6-point Likert scale. Response options range from 1 (strongly disagree) to 6 (strongly agree). Internal consistency scores diverge from high scores found in previous studies (Cronbach's α = .94), with a Cronbach's α = .61 in the present study.

**Sexual motives.** Sexual motivation was assessed using the sexual motives scale (SexMS) [43], a self-report questionnaire based on self-determination theory [42]. The SexMS consists of 24 items, with 4 items measuring each of six motivation styles: Amotivation, extrinsic motivation, introjected motivation, identified motivation, integrated motivation, and intrinsic motivation. Internal consistency showed to be good, with Cronbach's α above .80 on all subscales.

**The definition of sex and sexual agency.** To assess remaining psychosocial aspects that are not captured by the herein administered validated questionnaires (i.e., the definition of sex and sexual agency), a self-constructed questionnaire based on a number of questions presented in the Hite Report of Female Sexuality [47,56] was utilized. Coded qualitative responses published in the report were used as guidance for the presentation of response options. The questionnaire initially presented two descriptive items measuring the frequency of sexual activity and the frequency of how often sexual activity is desired, with response options ranging from 1 (more than once per day) to 8 (infrequently). Next, the participants' and the partners' definition of sex was measured by presenting various sexual behaviours that can be included or excluded into one's respective definition of "having sex". Further items measured the actual sexual behaviour including intimacy without PVI and the engagement in sexual behaviour solely to provide pleasure to one's partner or meet one's partner's expectations, desired sexual behaviour, sexual agency, emotional involvement during sexual activity, and importance and experience of orgasm. The questionnaire consisted of 46 items that could be responded to on a 5-point Likert scale ranging from 1 (completely disagree/ never) to 5 (strongly agree/ always). For the complete questionnaire, see the S1 File.

*Self-Constructed Scale Measuring Definition of Sex and Sexual Agency:*

| Dimension | Nr. of Items | Example Item |
|---|---|---|
| Descriptive Information | 3 | *Please indicate, how often you engage in sexual activity with your partner* |
| enm | 5 | *I engage in the following behaviours during sex** |
| Intimacy without PVI | 2 | *Physical touch and kissing without engaging in sexual intercourse is important to me* |
| Engagement to please partner or meet expectation | 5 | *I engage in the following behaviours **solely to meet my partner's expectations** (more than one option possible)** |
| Desired sexual behaviour | 5 | *I find the following behaviours pleasurable** |
| Sexual agency | 18 | *I feel guilty about taking time in sexual play that may not be stimulating to my partner** |
| Emotional involvement in sexual activity | 4 | *I am emotionally involved during sexual activity.* |
| Importance and experience of orgasm | 7 | *I experience orgasm during**. |

## Data reduction and development of self-constructed questionnaire scores

**Discrepancy between actual and desired sexual behaviour.** In order to derive insights into the discrepancy between actual and desired sexual behaviour, a difference score between the frequency of desired and the frequency of actual sexual behaviour was calculated (i.e., *desired frequency of sexual behaviour – actual frequency of sexual*

*behaviour*). Further, it was calculated whether a discrepancy exists between the sexual behaviours that are desired and those engaged in (i.e., *desired sexual behaviour – engaged in sexual behaviour*).

**Providing pleasure and meeting expectations.** To gain insights into the extent in which behaviours are engaged in solely to provide pleasure to one's partner or to meet one's partner's expectation, a mean score for each of the two respective items (*I engage in the following behaviours solely to provide pleasure to my partner; I engage in the following behaviours solely to meet my partner's expectation*) was calculated – the resulting scores are hereafter referred to as **provide pleasure** and **meet expectation**.

**Definition of sexual activity.** To gain insights into the sexual scripts and the broadness of the definition of sexual activity, a sum score and the mean for *PVI with* and *without orgasm* and for *all behaviours excluding PVI with and without orgasm* (oral sex, anal sex, kissing, masturbation, mutual masturbation, and intimacy and physical touch without penetration) were calculated for the participants and their partners. In a next step, a difference score was calculated between the agreement to PVI with and without orgasm as being part of the definition of sex and other behaviours as being defined as "having sex" for participants (*definition of sex PVI – definition of sex no PVI*), building a score named the **definition of sex**, of which higher scores indicate a more restricted definition of sex.

**Sexual agency.** To generate a measure for sexual agency, a principal component analysis of all items of the self-constructed "Definition of Sex and Aexual agency Questionnaire" related to the concept of sexual agency (defined as "the ability to identify, communicate, and negotiate one's sexual needs, and to initiate behaviours that allow for the satisfaction of those needs" [26]) was performed (see S2 File for more detailed information on the confirmatory factor analysis (CFA)). A total of 11 items was entered, and a varimax rotation was used, maximizing the variance of squared loadings within each factor, and yielding factors that are uncorrelated to each other. The CFA gave rise to four factors with eigenvalues above 1, of which the first two were entered into the heuristic model as theoretically relevant subscales of sexual agency. Factors 1 and 2 explain a cumulative variance of 42.0%.

**Factor 1.** Subscale 1 is comprised of the items *I am afraid to say no to sex, I engage in sexual activity I do not desire, I feel guilty about taking time in sexual play which may not be specifically stimulating to my partner,* and *I feel embarrassed asking for stimulation that does not specifically stimulate my partner*, which load strongly on Factor 1. Subscale 1 is named **Shame & Engagement in undesired sexual activity** and yields an internal consistency score of $\alpha = .76$.

**Factor 2.** The items *I engage in the following behaviours solely to provide pleasure to my partner* (provide pleasure) and *I engage in the following behaviours solely to meet my partner's expectation* (meet expectation) load strongly on Factor 2 and comprise Subscale 2, named **Partner's pleasure prioritization**. Subscale 2 yields an internal consistency score of $\alpha = .90$. The subscales are coded on a 5-point Likert scale from (0) to (4), in which high scores indicate high sexual agency, i.e., low partner's pleasure prioritization and low shame and engagement in undesired sexual activity.

## Statistical analysis

Following the descriptive analyses of vulvar pain and pain communication, as well as a thematic analysis of qualitative responses of reasons for not communicating pain, structural equation modelling (SEM) was used to test the relevance of the candidate factors as agents that either (i) directly influence sexual distress (FSDS - R total score), sexual function (FSFI total score), and vulvar pain, or ii) indirectly via engagement in sexual intercourse despite pain, sexual pain communication, relationship satisfaction, and dyadic sexual communication. Sexual self-esteem, the definition of sex, shame & engagement in undesired sexual activity (sexual agency subscale 1), partner's pleasure prioritization (sexual agency subscale 2), and sexual motivation were included in the model as candidate causal agents. Due to normal distribution of the data, maximum likelihood parameter estimation was chosen [57]. The predictors were further tested for multicollinearity and were allowed to covary in the present model. Assumptions of multivariate normality and linearity were evaluated with SPSS Version 29. Fig 1 depicts the hypothesized model that was taken as starting point.

## Results

### Descriptive statistics

The present sample had a mean FSDS score of $M = 13.8$ ($SD = 11.25$), with 51.2% of women scoring above the cut-off score of 11, indicating sexual distress. The mean FSFI score was $M = 23.76$ ($SD = 8.9$), with 48.3% of women below the cutoff score of 26, indicating female sexual dysfunction. For indications of orgasm frequency, see Table 1.

### Vulvar pain and pain communication

When asked about the presence of physical discomfort or pain during attempted or engagement in PVI with their partners, 75.9% of women reported experiencing pain *at least sometimes*. Furthermore, 15.7% of women indicated to experience pain *more than half of the time* during (attempted) PVI (see Table 2). Factors contributing to pain included extended or frequent sexual activity (29.7%), insufficient arousal or lubrication (51.3%), deep thrusts (23.7%), a partner with a too-large penis (17.2%), discomfort at the beginning of intercourse (16.4%), and yeast infections (8.2%). Additional factors given by participants included body tension, vaginismus, dyspareunia, presence of an intrauterine device, sexual trauma, and stress. The distribution of average pain level ratings in percentages is illustrated in Fig 2.

When asked whether participants had engaged in sexual activity despite pain, 65.4% of women in the present sample *at least sometimes* engaged in sexual intercourse despite pain, while 45.8% of women *most of the time* or *always* engaged in intercourse despite pain (see Table 2). Regarding whether affected women had informed their partner about being in pain during sexual activity (in case their most recent sexual activity was painful, which was the case in 53% of women), 53.4% indicated that they had communicated their pain, while 41.2% had not done so. A thematic analysis of presented reasons for not communicating the pain unveiled recurrent themes, including *fear of disrupting the moment, concerns about the partner's feelings, social pressure and embarrassment, initial pain that fades eventually and is not experienced as too painful (or even as pleasurable), and avoidance of awkward conversations or situations/shame* (see Table 3).

### Factors explaining vulvar pain during PVI and engagement in painful sexual behaviour amidst pain

The hypothesized model ($\chi^2$ (4), N = 232 = 10.688, p < .031) included direct and indirect effects of all dependent variables and mediating variables to all outcome variables (Fig 3). Missing data was imputed based on regression imputation

**Table 1. Orgasm frequency in percent in masturbation and partnered sex.**

|  | Masturbation | Partnered Sex |
|---|---|---|
| Never | 8.1% | 10.8% |
| Rarely | 5.9% | 19.8% |
| Sometimes | 9.5% | 29.7% |
| Often | 28.4% | 33.8% |
| Always | 48.2% | 5.9% |

**Table 2. Percentage of women experiencing pain during PVI and engaging in PVI despite pain on a scale from always to never.**

|  | Yes, always | Yes, most of the times | Yes, more than half of the times | Yes, sometimes | No, never |
|---|---|---|---|---|---|
| Experience of Pain During PVI | 1.4% | 6.5% | 7.9% | 60.2% | 24.1% |
|  | Yes, always | Yes, most of the times | Yes, sometimes | Yes, but rarely | No |
| Engagement in PVI despite pain | 15.0% | 30.8% | 19.6% | 15.0% | 19.6% |

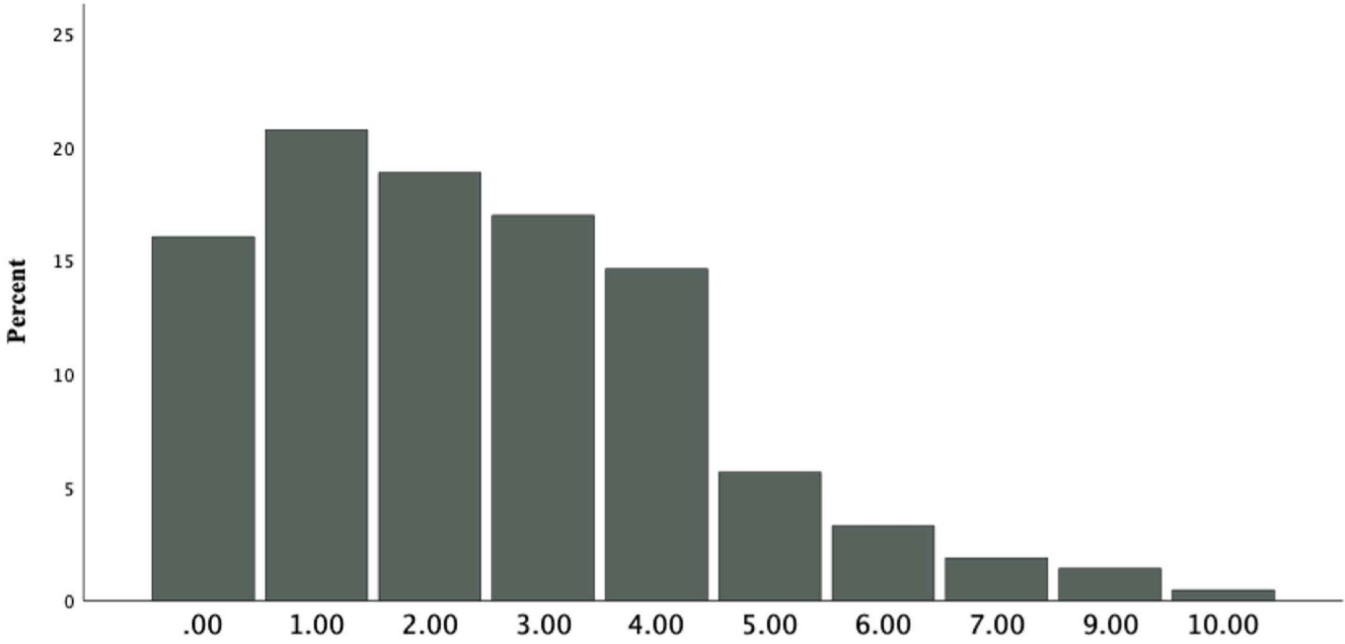

**Fig 2. Percentage of average pain levels on a scale from 1 (no pain) to 10 (worst pain).**

and bootstrapping was performed as recommended by Preacher and Hayes [58], estimating 2000 resamples at a bias-corrected confidence level of 90. As can be seen in Table 4, model fit indices improved once dyadic sexual communication was removed as a mediator. As predictors were allowed to covary in the present model, their covariances and correlations can be seen in S1 Table.

Due to good model fit, no further post-hoc modifications were conducted. The standardized and unstandardized coefficients of the final model are depicted in S2 Table. To improve legibility, a parsimonious model including main effects (red) and mediation effects (black) is depicted in Fig 2.

### Factors influencing women's persistence in sexual activity amidst painful experiences and pain communication

Engagement in sexual intercourse despite pain was increased in individuals with higher scores on shame and engagement in undesired sexual activity, while higher autonomous sexual motivation was associated with significantly reduced engagement in sexual intercourse despite pain. Whether women communicated their pain to their partners was associated with the degree of partner's pleasure prioritization, as higher pleasure prioritization predicted reduced communication of pain, and the definition of sex. Higher sexual self-esteem predicted higher relationship satisfaction, which was related to reduced sexual distress, and increased sexual function.

### Factors influencing sexual distress, vulvar pain, and sexual functioning

**Sexual distress.** A restricted definition of sex, i.e., holding a definition of sex limited to penetrative sexual intercourse, was associated with increased sexual distress whereas sexual self-esteem, relationship satisfaction, pain communication and both dimensions of sexual agency were associated with reduced sexual distress. The effect of sexual self-esteem on sexual distress was (partly) mediated by relationship satisfaction. The effects of definition of sex and partner pleasure prioritization and sexual distress were mediated by pain communication.

**Table 3. Reasons for not communicating pain during sexual activity.**

| Theme | Example |
|---|---|
| Fear of disrupting the moment | *I did not want to interrupt the moment* <br> *I did not want to bother him and make it a thing* <br> *I did not want to ruin the moment or the experience for him* <br> *It will ruin the mood* |
| Concern about Partner's feelings | *I did not want him to feel ashamed* <br> *I did not want to make him uncomfortable* <br> *I did not want to disappoint them* |
| Social pressure, shame and guilt | *I felt guilty for stopping in the middle of having sex,* <br> *I thought it was easier to wait until he experiences an orgasm* <br> *I was embarrassed* <br> *I was a bit ashamed* |
| Initial pain that fades or is experienced as pleasurable | *Because it usually gets pleasurable after some time* <br> *Because it is usually just in the beginning* |
| Avoidance of awkward situations or conversations | *I did not want to make it awkward* <br> *I did not want to bother* <br> *To avoid making a scene* |

**Vulvar pain.** A restrictive definition of sex (i.e., holding a definition of sex limited to penetrative sexual intercourse) was associated with increased vulvar pain, sexual self-esteem, pain communication, and engagement in PVI despite pain were associated with reduced vulvar pain. The effect of definition of sex on vulvar pain was mediated by pain communication. The effects of sexual agency subscale 1 and autonomous sexual motivation were mediated by engagement in sexual intercourse despite pain. The effect of sexual agency subscale 2 on vulvar pain was mediated by pain communication.

**Sexual function.** Autonomous sexual motivation, sexual agency subscale 1, and relationship satisfaction were associated with increased sexual function.

## Discussion

The present study showed that the experience of vulvar pain during sexual intercourse continues to be prevalent in the female population, with 80.0% of women reporting to experience pain at least sometimes, and 15.0% reporting to experience pain more than half of the time. About half of the women scored within ranges of sexual distress and sexual dysfunction, indicating a high prevalence of sexual difficulties. Engaging in sexual intercourse despite experiencing pain was common, with 42.0% of women indicating to do so always or most of the time and 65.0% at least sometimes when experiencing pain. Of the affected women, 41.0% did not communicate their pain to their partners, bringing forward reasons such as fear of disrupting the moment, concerns about the partner's feelings, social pressure and embarrassment, and avoidance of awkward conversations or situations and shame. Sexual distress, vulvar pain, and sexual function were found to be associated with the psychosocial factors that were included in the model. Precisely, sexual self-esteem and definition of sex were associated with sexual distress and vulvar pain, both sexual agency subscales were associated with sexual distress, partner pleasure prioritization (sexual agency subscale 2) was associated with vulvar pain and sexual function, and autonomous sexual motivation was associated with sexual function. These relationships were partly mediated by relationship satisfaction, sexual pain communication, and engagement in sexual intercourse despite pain.

The present results correspond to previous research finding that about 50.0% of women do not communicate pain to their partners, for reasons such as the normalisation of painful sex and avoiding uncomfortable situations, prioritising the partner's pleasure, shame and embarrassment, and for promoting the relationship [6], as well as to the

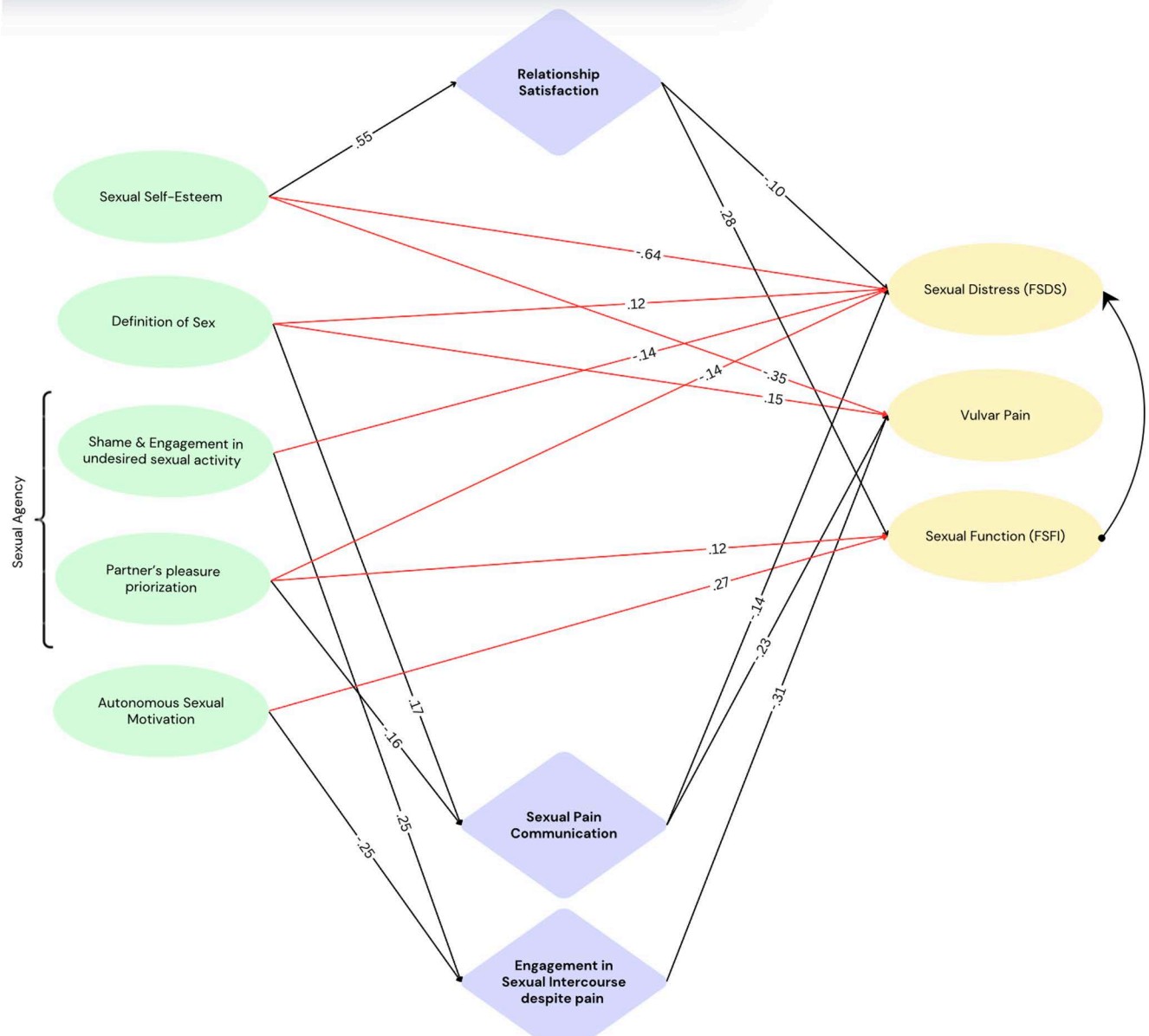

**Fig 3. SEM Parsimonious Model.** Note. Green: Observed predictor variables (model allows for covariance among predictor variables). Lilac: Observed mediator variables. Yellow: Observed outcome variables. Red arrows: Direct effects. Black arrows: Mediation effects.

**Table 4. Model Fit Indices.**

| Model Fit Indices | Model 1 (Hypothesized Model) | Model 2 (Final Model) |
|---|---|---|
| CFI | .967 | .989 |
| TLI | .744 | .855 |
| RMSEA | .085 | .085 |

previously postulated notion that engagement in sexual activity despite pain is common among women [4,5]. The heuristic model (Fig 2) extends previous research by providing insights into psychosocial factors that may play a relevant role in motivating engagement in sexual intercourse despite pain and reluctance of pain communication. While autonomous sexual motivation was associated with reduced engagement in sexual intercourse despite pain, shame and engagement in undesired sexual activity (sexual agency subscale 1) showed a positive association. Further, women with a strong tendency to prioritize their partner's pleasure (sexual agency subscale 2), showed reduced communication of pain, while a restrictive definition of sex was positively associated with pain communication – this may result from a restrictive definition of sex being related to increased vulvar pain and sexual distress, heightening the need to communicating pain as alternative intimate activities are not assigned with the same value as PVI. Accordingly, both relationships between a restrictive definition of sex and vulvar pain and sexual distress were mediated by pain communication, indicating that a restrictive definition of sex increases vulvar pain and distress, which is further augmented if pain is not communicated.

In line with expectations, each of the psychosocial factors included into the model was associated with at least one of the outcome variables – sexual distress, vulvar pain, and sexual function. As hypothesized, both sexual agency subscales were associated with reduced sexual distress. This finding demonstrates that fostering a feeling of agency and autonomy may lead to sexual behaviour more in line with desires and own preferences and prevent engagement in activities that elicit pain and discomfort, germane to the finding that partner's pleasure prioritization is associated to reduced pain communication, which is associated to increased distress. Being able to communicate sexual needs and find a balance between the partner's and one's own pleasure may thus limit sexual distress. Furthermore, body-image and overall confidence and self-esteem may also play an explanatory role in the association between sexual agency and distress, as women with higher self-esteem and confidence may find it easier to act sexually agent and may be less prone to experience high sexual distress in painful or undesired sexual situations [59]. Furthermore, communication of sexual needs may come easier [39].

Sexual self-esteem was positively related to relationship satisfaction and showed a negative association with sexual distress and vulvar pain, indicating reduced sexual distress and pain in individuals with higher sexual self-esteem. This is in line with research finding higher SSE levels to be associated with reduced psychological distress, while decreased SSE among correlated with elevated levels of depression, anxiety, and stress in menopausal women [60], and difficulties in sexual and marital relationships [61]. However, sexual intercourse despite pain and pain communication did not mediate the relationships between sexual self-esteem and sexual distress or vulvar pain. In line with predictions, autonomous sexual motivation had a positive effect on sexual function and was negatively associated with engagement in sexual intercourse despite pain.

## Clinical implications and strengths

With about half of the present sample exceeding the cut-off scores for sexual dysfunction and distress, the need for investigating causal and maintaining factors becomes apparent. The present results highlight the importance of investigating psychosocial concepts such as sexual self-esteem, sexual agency, and sexual motivation in relation to vulvar pain and sexual distress in women. Furthermore, the present study points to the relevance of establishing a broader definition of sex that is not limited to PVI to reduce sexual distress and vulvar pain in women.

The findings provide a heuristic model that may serve as theoretical underpinning and guideline for interventions for women experiencing vulvar pain and sexual distress or reduced sexual function and their partners, and points to psychosocial factors that likely play a role in the cause and/or maintenance of sexual pain and should be considered besides biological factors. The model emphasizes the importance of targeting sexual self-esteem and sexual agency in order to prevent women engaging in sexual activities that cause pain and to foster communication of pain to their partners and improve relationship satisfaction.

Further, the present study shows that acting sexually agent and discordant to traditional gender roles can be associated with sexual distress, positing one reason why women may prefer prioritizing the partner's pleasure over their own, possibly finding pleasure in the partner's pleasure. It is important to clarify that this is not necessarily negative, as long as the partner's pleasure does not come at costs of the participant's own sexual pleasure or health. Germane to previous research, the present study found women to frequently refrain from communicating pain in order to prevent awkwardness and embarrassment or shame and provide pleasure to the partner and that rates of engagement in PVI despite pain are high [6]. The high numbers of women experiencing sexual distress and reduced sexual function, hold a restricted definition of sex, and/or engage in painful sex in a "healthy" student sample also highlights the deep embeddedness of sexual scripts and social norms, that may be learned from early on and may be difficult to overcome [14]. This seems to hold in spite the general movement towards challenging gender norms and the affirmation of (sexual) empowerment and agency of women and young adults [19,62], and the increasing importance given to sexual pleasure [63–65]. Therefore, the present results corroborate the idea that promoting women's own satisfaction and engagement in intimate activities beyond PVI remains necessary [18,29].

Lastly, the present study provides a novel approach to measuring sexual agency by measuring degrees of shame and guilt when asking for stimulation only pleasurable or stimulating to the women themselves, the ability to say no, and engagement in undesired sexual activities. While this measurement is preliminary and remains to be validated in a different and independent sample, it is a first step towards a methodologically sound measurement tool of sexual agency, which is yet to be developed. This study replicates findings from Carter et al. [6] and extends existing research by introducing the definition of sex as a relevant factor. It underscores the importance of sexual agency, sexual self-esteem, and sexual motivation in addressing sexual function, distress, vulvar pain, engagement in sexual intercourse despite pain, and communication of pain.

## Limitations and future directions

The present study is not free of limitations and methodological shortcomings that warrant consideration. A wide array of psychosocial factors have been related to vulvar pain and sexual distress [1]. Future research may look into the role of anxiety and depression [66–68], attachment style [69], history of abuse and pain catastrophizing [1,3], body image [70–73], and overall self-esteem [74,75] in the context of the psychosocial factors tested in the herein presented model. This study comprised a homogenous sample of female university students, of which the majority was heterosexual or indicated to engage in sexual activities with men and had a western and educated background. While young women comprised a valuable starting point to text the hypothesized model, the homogenous sample renders the study vulnerable to selection bias. In a next step, it would therefore be important for future research to test whether the model generalizes to larger, more diverse samples. Actively sampling women who engage in sexual activity with women may be of particular value, as there may be less pressure to engage in penetrative intercourse and the definition of sex is potentially wider and less scripted. Further, the sample was, on average, very young, which may have resulted in an increased prevalence of vulvar pain compared to older women women [76]. As prevalence of vulvar pain seems to decrease with age, including older women may be relevant to derive inferences on whether sexual agency increases, while engagement in unwanted or painful sexual behaviour decreases with age and whether the relevant psychosocial factors change across different periods in women's life, for example before and after childbirth or post menopause. Next, the sexual agency questionnaire was developed for this study and requires further validation. Due to the limited sample size, the sum-scores of validated questionnaires were used instead of entering all variables as indicator variables loading on a latent construct. This may have resulted in an underestimation of measurement error. Lastly, SEM was used within the context of an inductive and data-driven approach to develop the model of best fit to represent the data. It is therefore important for future research to test the robustness of the final model in an independent sample. Future samples could extent to women affected by chronic health conditions such as mental health conditions, autoimmune or inflammatory diseases, chronic pain, or

post-cancer (treatment). Women with a sexual dysfunction diagnosis or in a relationship with a partner diagnosed with sexual dysfunction could also be included to gain insights into the role of sexual agency and sexual communication in the development of secondary sexual dysfunction.

## Conclusion

The present study found sexual distress and limited sexual function to be high in female university students, in line with large numbers engaging in sexual intercourse despite pain and not communicating their pain to their partners. The herein presented heuristic model suggests that sexual self-esteem, sexual agency, the definition of sex, and sexual motivation warrant consideration when considering possible causal and maintaining factors of vulvar pain and sexual distress. The fact that a restrictive definition of sex was associated with sexual distress and vulvar pain is consistent with the notion that engagement in intimacy and behaviours besides PVI should be promoted in order to move away from PVI being the epitome of "having sex" [16,17]. Further, findings point to the importance of promoting female entitlement to pleasure and sexual agency, autonomous sexual behaviour, and sexual self-esteem, which may also result in increased relationship satisfaction and overall well-being by reducing sexual distress and dysfunction.

## Supporting information

**S1 File. Complete questionnaire.**
(DOCX)

**S2 File. Confirmatory Factor Analysis.**
(DOCX)

**S1 Table. Covariances and correlations of predictor variables.**
(DOCX)

**S2 Table. Standardized and unstandardized coefficients of final model.**
(DOCX)

## Acknowledgments

The Authors thank M.Sc. students Kim Walk, Ava Tauser, and Carol Gontijo-Santos Lima, as well as B.Sc. students Julia Linnemann and Kira Krebs for their contribution to the design and implementation of the study, as well as to the data collection as partial fulfilment of their theses. Moreover, the authors thank Prof. Dr. Mark Huisman and Prof. Dr. Jeffrey Kiesner for their statistical guidance in structural equation modelling and Dr. Paula von Spreckelsen and Piet van Tuijl for their advice regarding confirmatory factor analysis. The authors thank all participants for their participation, making this study possible.

## Author contributions

**Conceptualization:** Carlotta Oesterling, Amelie Harder, Charmaine Borg, Peter de Jong.

**Data curation:** Carlotta Oesterling, Amelie Harder, Charmaine Borg, Peter de Jong.

**Formal analysis:** Carlotta Oesterling.

**Investigation:** Carlotta Oesterling, Charmaine Borg, Peter de Jong.

**Methodology:** Carlotta Oesterling, Charmaine Borg, Peter de Jong.

**Project administration:** Carlotta Oesterling, Charmaine Borg, Peter de Jong.

**Resources:** Carlotta Oesterling, Amelie Harder, Charmaine Borg, Peter de Jong.

**Supervision:** Charmaine Borg, Peter de Jong.

**Visualization:** Carlotta Oesterling.

**Writing – original draft:** Carlotta Oesterling.

**Writing – review & editing:** Carlotta Oesterling, Charmaine Borg, Peter de Jong.

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
