## [Decision Letter · Decision Letter 0]

5 Nov 2024

PONE-D-24-21802Factors involved in vulvar pain during sexual activity and persistence in sexual activity amidst painful experiences.PLOS ONE

Dear Dr. Oesterling,

Thank you for submitting your manuscript to PLOS ONE. After careful consideration, we feel that it has merit but does not fully meet PLOS ONE’s publication criteria as it currently stands. Therefore, we invite you to submit a revised version of the manuscript that addresses the points raised during the review process.

Please answer all reviewer comments, even those by reviewer #1

We look forward to receiving your revised manuscript.

Kind regards,

Marta Panzeri, Ph.D.

Academic Editor

PLOS ONE

Journal requirements:  When submitting your revision, we need you to address these additional requirements.   1. Please ensure that your manuscript meets PLOS ONE's style requirements, including those for file naming. The PLOS ONE style templates can be found at  https://journals.plos.org/plosone/s/file?id=wjVg/PLOSOne_formatting_sample_main_body.pdf and  https://journals.plos.org/plosone/s/file?id=ba62/PLOSOne_formatting_sample_title_authors_affiliations.pdf  2. When completing the data availability statement of the submission form, you indicated that you will make your data available on acceptance. We strongly recommend all authors decide on a data sharing plan before acceptance, as the process can be lengthy and hold up publication timelines. Please note that, though access restrictions are acceptable now, your entire data will need to be made freely accessible if your manuscript is accepted for publication. This policy applies to all data except where public deposition would breach compliance with the protocol approved by your research ethics board. If you are unable to adhere to our open data policy, please kindly revise your statement to explain your reasoning and we will seek the editor's input on an exemption. Please be assured that, once you have provided your new statement, the assessment of your exemption will not hold up the peer review process. 3. Please amend either the title on the online submission form (via Edit Submission) or the title in the manuscript so that they are identical.

Reviewers' comments:

Reviewer's Responses to Questions

**Comments to the Author**

1. Is the manuscript technically sound, and do the data support the conclusions?

Reviewer #1: Yes

Reviewer #2: Yes

2. Has the statistical analysis been performed appropriately and rigorously? 

Reviewer #1: Yes

Reviewer #2: Yes

3. Have the authors made all data underlying the findings in their manuscript fully available?

Reviewer #1: Yes

Reviewer #2: Yes

4. Is the manuscript presented in an intelligible fashion and written in standard English?

Reviewer #1: Yes

Reviewer #2: Yes

5. Review Comments to the Author

Reviewer #1: MINOR REVISION

- place the young average age within the limits and must take this into account in the discussion

- put the description of the self-constructed questionnaire in the tools section and not in the statistical analysis section

- always use the same number of decimals in percentages

Reviewer #2: This study tries to evaluate which psycho-social factors may explain the link between the presence of genital pain and the willingness to have penetrative sexual intercourse in heterosexual women. The authors reported interesting data. First, the presence of a very high prevalence of genital pain (80%) in a general population. Yet this information should be taken in mind and deserves a broader reflection on the female sexuality. Then the authors found a significant association between sexual distress, vulvar pain, and sexual function, with some other psychosocial variables: sexual self-esteem, definition of sex, sexual distress, sexual agency ,autonomous sexual motivation. These relationships were mediated by relationship satisfaction, sexual pain communication, and engagement in sexual intercourse despite pain. I really appreciate this work, which gives to the readers a broader picture of the reasons why women experiencing pain engage in penetrative sex . I think that this work can be very useful from clinical point of view. Here I listed some minor comments.

Introduction

If I correctly understand, this study is aiming at evaluating general painful sensations in relationship to sexual activity or intercourse. For this reason, I suggest adopting the name of “genital pain” rather than of “vulvar pain” (as adopted by the authors) through the manuscript.

Where the authors mention the types of vulvar pain, indicating the “idea of penetration”, I suggest adding the possible diagnosis of vaginismus.

The authors refer to PVI of heterosexual women. It could be also useful to add the information of a lack of data on non-heterosexual couples.

From my point of view, the introduction is too long. I suggest summarizing the central points related to the main topic.

Methods:

The authors have to explain how they structured the items evaluating vulvar pain and sexual intercourse.

It could be useful to add the information about the type of diagnosis related to genital pain and its prevalence. Another important variable is the type of treatment and its efficacy in acting on pain management.

Definition of sexual agency: I don’t understand if the authors adopted only a limited number of the items present in the Hite Report of Female Sexuality.

In the results, I suggest to better define the “restrictive definition of sex”.

I suggest reporting, in the text or with a table, the prevalence of sexual dysfunctions. I think that this data

6. PLOS authors have the option to publish the peer review history of their article (what does this mean? ). If published, this will include your full peer review and any attached files.

**Do you want your identity to be public for this peer review?** For information about this choice, including consent withdrawal, please see our Privacy Policy .

Reviewer #1: No

Reviewer #2: No

---

## [Author Response · Author response to Decision Letter 1]

10 Jan 2025

Thank you very much for the positive feedback and the appreciation of our work. We have found the detailed suggestions and textual recommendations very helpful, and thus we have implemented them meticulously, hoping that sections that were somewhat unclear are clearer now. In a seperate document that can be found in the attached files, we have responded to each comment individually. We are very much looking forward to hearing your feedback.

---

## [Decision Letter · Decision Letter 1]

4 Feb 2025

PONE-D-24-21802R1Factors Involved in Vulvar Pain during Sexual Activity and Persistence in Sexual Activity amidst PainPLOS ONE

Dear Dr. Oesterling,

Thank you for submitting your manuscript to PLOS ONE. After careful consideration, we feel that it has merit but does not fully meet PLOS ONE’s publication criteria as it currently stands. Therefore, we invite you to submit a revised version of the manuscript that addresses the points raised during the review process.

Could you please address all the reviewer comments and write a rebuttal letter that addresses each point, rather than just a general sentence for all your revisions?

I would also like to mention that review #2 was intended to select minor revisions for the paper.

We look forward to receiving your revised manuscript.

Kind regards,

Marta Panzeri, Ph.D.

Academic Editor

PLOS ONE

Journal Requirements:

Additional Editor Comments:

Could you please address all the reviewer comments and write a rebuttal letter that addresses each point, rather than just a general sentence for all your revisions?

I would also like to mention that review #2 was intended to select minor revisions for the paper.

Reviewers' comments:

Reviewer's Responses to Questions

**Comments to the Author**

1. If the authors have adequately addressed your comments raised in a previous round of review and you feel that this manuscript is now acceptable for publication, you may indicate that here to bypass the “Comments to the Author” section, enter your conflict of interest statement in the “Confidential to Editor” section, and submit your "Accept" recommendation.

Reviewer #2: (No Response)

2. Is the manuscript technically sound, and do the data support the conclusions?

Reviewer #2: Yes

3. Has the statistical analysis been performed appropriately and rigorously? 

Reviewer #2: Yes

4. Have the authors made all data underlying the findings in their manuscript fully available?

Reviewer #2: Yes

5. Is the manuscript presented in an intelligible fashion and written in standard English?

Reviewer #2: Yes

6. Review Comments to the Author

Reviewer #2: This study will evaluate, in a population of Dutch students, which psycho-social factors are involved in the painful sexual activity, and in the lack of pain-communication to the partner. The Authors find a very high percentage of young women referring to experience pain during sexual intercourse. In addition, female distress, sexual function and vulvar pain were associated with low sexual self-esteem, low self-motivation for sex, and limited sexual agency. I appreciated this work, even if some concerns are present. Here I listed my comments and suggestions.

How the authors explain the very high percentage of vulvar pain in their sample, compared with the literature data? Have the authors considered to go in depth with the analysis of the prevalence, considering separate sub-groups pf women? I think that sexual experience may profoundly change in 18- or 30-years women.

The authors must explain why they decided to measure sexual motives adopting the construct of sexual scripts.

The part dedicated to the description of sexual scripts should be shortened and more centered on the main topic of the study.

I would like to know if the authors have assesses in the first part of the study the presence of diagnosed sexual dysfunctions, of organic conditions suggesting the presence of pain, or of psychopathological traits/psychiatric diagnoses associated with vulvar pain.

I think that an assessment of partner’s sexual health could be important to detect.

The enrollment of an exclusive population of young women should represent a selection bias. I suggest adding this point to the limits.

I don’t understand if vulvar pain assessment is made with no-validated items. If yes, the authors are invited to explain the methodology for selecting and structuring these questions.

Pleasure evaluation: the authors propose to the subjects the following two questions:” I engage in the following behaviours solely to provide pleasure to my partner; I engage in the following behaviours solely to meet my partner's expectation.” From my point of view, an additional question matching together the two options (self- and partner- pleasure) is necessary.

7. PLOS authors have the option to publish the peer review history of their article (what does this mean? ). If published, this will include your full peer review and any attached files.

**Do you want your identity to be public for this peer review?** For information about this choice, including consent withdrawal, please see our Privacy Policy .

Reviewer #2: No

---

## [Author Response · Author response to Decision Letter 2]

19 Mar 2025

Dear Dr. Marta Panzeri, Dear Reviewers,

Hereby we would like to thank you and the reviewer for the additional time dedicated to our manuscript (PONE-D-24- 21802), entitled "Factors involved in vulvar pain during sexual activity and persistence in sexual activity amidst painful experiences."

We gladly updated our manuscript in line with the reviewer’s comments and have formulated a rebuttal letter to address each comment individually (see attached document). The constructive comments and points addressed by the reviewer are highly appreciated. We hope the remaining questions are now addressed and that the revised manuscript now fully qualifies for publication in PLOS ONE.

With warm regards,

Carlotta Oesterling,

on behalf of the co-authors.

---

## [Editor Report · Decision Letter 2]

15 Apr 2025

Factors Involved in Vulvar Pain during Sexual Activity and Persistence in Sexual Activity amidst Pain

PONE-D-24-21802R2

Dear Dr. Oesterling,

We’re pleased to inform you that your manuscript has been judged scientifically suitable for publication and will be formally accepted for publication once it meets all outstanding technical requirements.

Kind regards,

Marta Panzeri, Ph.D.

Academic Editor

PLOS ONE
---

## [Editor Report · Acceptance letter]

PONE-D-24-21802R2

PLOS ONE

Dear Dr. Oesterling,

I'm pleased to inform you that your manuscript has been deemed suitable for publication in PLOS ONE. Congratulations! Your manuscript is now being handed over to our production team.

Kind regards,

on behalf of

Dr. Marta Panzeri

Academic Editor

PLOS ONE